# Exploring Minimally Sufficient Representation in Active Learning through Label-Irrelevant Patch Augmentation

Zhiyu Xue[1]      Yinlong Dai[2]      Qi Lei[2]
[1]UC Santa Barbara, [2]New York University
zhiyuxue@ucsb.edu, {yd2032,ql518}@nyu.edu

Deep learning models, which require abundant labeled data for training, are expensive and time-consuming to implement, particularly in medical imaging. Active learning (AL) aims to maximize model performance with few labeled samples by gradually expanding and labeling a new training set. In this work, we intend to learn a "good" feature representation that is both sufficient and minimal, facilitating effective AL for medical image classification. This work proposes an efficient AL framework based on off-the-shelf self-supervised learning models, complemented by a label-irrelevant patch augmentation scheme. This scheme is designed to reduce redundancy in the learned features and mitigate overfitting in the progress of AL. Our framework offers efficiency to AL in terms of parameters, samples, and computational costs. The benefits of this approach are extensively validated across various medical image classification tasks employing different AL strategies. [1].

## 1. Introduction

Deep learning models typically require training with abundant labeled data. However, annotating medical images requires prior domain expertise and is both costly and time-consuming. A potential mitigation for this challenge is through active learning (AL). AL aims to optimize model performance using the smallest number of labeled samples possible by incrementally expanding and labeling the training set. By prioritizing labeling informative samples rather than random selections, AL significantly enhances sample efficiency [1].

Recent advances in AL largely attribute to the development of modern deep learning models (See reference therein [2–8]). Given that large-scale deep learning models are even more sample-demanding, it is urgent to develop effective and

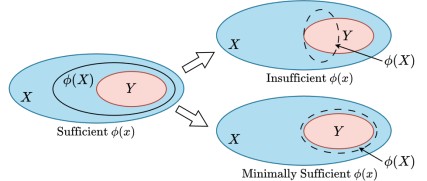

Figure 1: Information diagram for insufficient, sufficient, and minimally sufficient $\phi(x)$. Ideally, AL can lead sufficient $\phi(x)$ to be gradually closer to minimally sufficient $\phi(x)$. However, the lack of labeled samples can also result in an insufficient $\phi(x)$.

efficient active learning strategies. These strategies are essential to minimize the opportunity cost of labeling redundant samples, a significant concern in medical image classification where human annotations are notably scarce and costly.

In this work, we argue that the key to successful AL is to learn a "good" feature representation $\phi : x \to \phi(x)$. Ideally, this representation should ensure that label $y$ is linearly separable in the representation space. Such representation is preferable, since the process of AL for linearly separable data is well-understood [1, 9]. As depicted in Fig. 1, establishing such a good representation requires the following conditions: (**a**) the representation we learned should be **sufficient** to predict $y$. This means $\phi(x)$ does not lose essential features in $x$ that are relevant to $y$. Mathematically speaking, our objective is to ensure $P(y|x)$ in the classification task aligns with $P(y|\phi(x))$, therefore constraining the

---

[1]Source Codes: https://github.com/chrisyxue/DA4AL

First Conference on Parsimony and Learning (CPAL 2024).

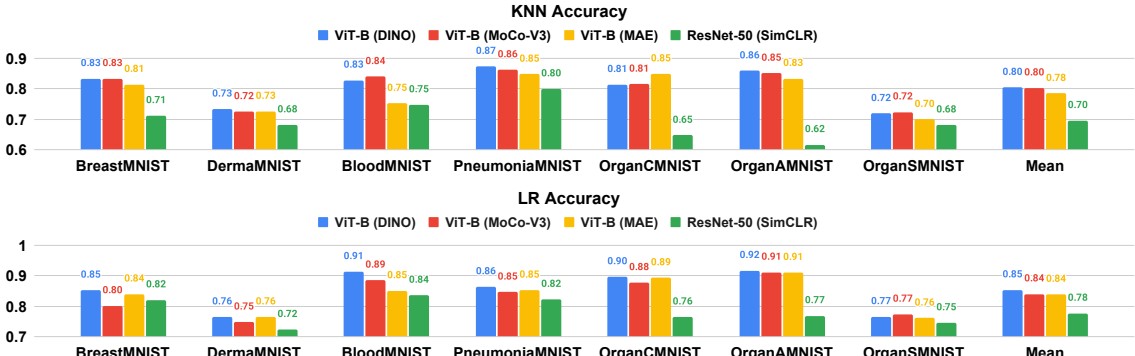

Figure 2: To evaluate the quality of representations $\phi(x)$ for off-the-shelf SSL models in downstream tasks, we employ k-nearest neighbors (KNN) and logistic regression (LR) to model $P(y|\phi(x))$ based on the off-the-shelf SSL models including ViT-B with checkpoints released by DINO [14], MoCo-V3 [15], and MAE [16], and ResNet50 with checkpoint released by SimCLR [17]. The accuracy of these classifiers serves as our metric for assessing the linear separability of the representations produced by these off-the-shelf SSL models.

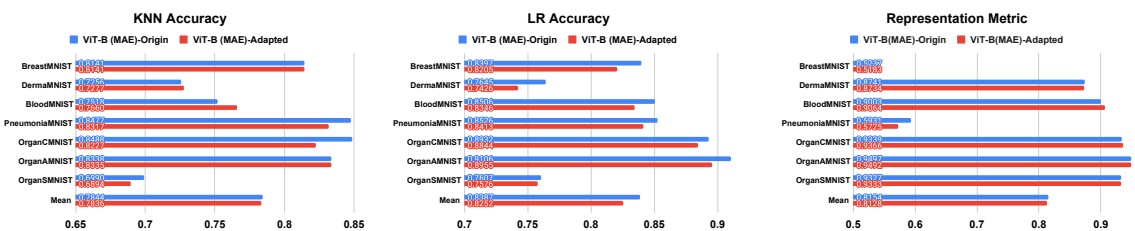

Figure 3: We compared the representation quality of adapting or not adapting the off-the-shelf ViT-B (MAE) to the unlabeled downstream tasks of medical images by reconstruction-based pretext training [16], where **ViT-B (MAE)-Adapted**/**ViT-B (MAE)-Origin** indicates the results of adapted/unadapted off-the-shelf ViTs gaining from the checkpoints released by MAE [16], respectively. Results show that **ViT-B (MAE)-Origin** can most times gain better representations than **ViT-B (MAE)-Adapted**.

classifier from a function on $x$ to a function on $\phi(x)$ introduces no additional bias. However, merely fulfilling this condition does not guarantee tangible benefits. For instance, a naive choice of sufficient $\phi$ such as identity mapping fails in feature reduction. Therefore another essential requirement is (**b**) the representation should be **minimal**, in the sense that it preserves only the crucial information necessary to predict $y$. By constraining the classifier to take in $\phi(x)$ instead of $x$, assuming $\phi(x)$ excludes redundant features of $x$, the model becomes more sample-efficient and will generalize better. The less redundant information $\phi(x)$ contains, given its sufficiency, the more sample-efficient it is to model the relationship between $\phi(x)$ to $y$ [10, 11]. This concept will be especially beneficial in the few-shot regime, aligned with the purpose of AL. (**c**) In line with AL's principles, this representation initially can only be **approximated by off-the-shelf models** that have been pretrained on large-scale tasks with self-supervised learning (SSL) strategies, and gradually be corrected/fine-tuned with the progress of labeling new training samples from downstream tasks. Integrating conditions (a)(b) and (c), as proven in the theoretical work [12, 13] for SSL, an approximately minimally sufficient representation established from the large-scale SSL tasks ensures our targets to be linearly separable in the representation space for downstream tasks.

We investigate practical strategies to fulfill the above conditions and together propose an effective, sample- and computationally efficient AL framework.

First, as argued above, we employ off-the-shelf SSL models to gain approximately minimally sufficient representations for medical images. Specifically, we choose off-the-shelf ViTs (checkpoints as DINO [14], MoCo-V3 [15], and MAE [16]) as the backbone initially for AL.

Compared to other architectures for off-the-shelf SSL models like ResNets [18] (checkpoint as SimCLR [17]), off-the-shelf ViTs explicitly contain detailed information like object shapes and textures [14] which are essential for analyzing medical images; we also show in Fig. 2 that they have great potential to achieve linearly separable representations for medical images across different diseases and modalities. Besides, we surprisingly find that adapting off-the-shelf ViTs to the unlabeled downstream medical image tasks by pretext training can reduce but cannot improve the representation quality (Fig. 3). Therefore, utilizing off-the-shelf ViTs directly to AL on the downstream tasks of medical images is both an effective and efficient choice.

With the increasing labeling information in AL, we need to accordingly design effective algorithms to improve the feature representation (to gradually be closer to being minimally sufficient). This is challenging since labeled samples are very limited, and thus it is unrealistic to fine-tune the whole representation, with risks of distorting the originally good features and overfitting the little samples [19–21]. We resolve this problem by proposing label-irrelevant patch augmentations and by learning only a subsequent layer as an adapter on top of the fixed pre-trained representation. By investigating a diversity of label-irrelevant patches, we largely enrich the training set and ameliorate the overfitting issue in the early stages of AL. Unlike traditional data augmentations [22–24] that tend to modify the semantics in medical domains [25, 26] and cause additional errors due to misspecifications, our method utilizes the little labeling information to guarantee no semantics is changed in the augmented data, leading to more reliable and robust feature learning.

Based on our proposed framework above, our contributions can be concluded as follows:

- We design a parameter-, sample- and computationally efficient AL framework based on self-supervised pretrained ViTs, to initially gain nearly minimally sufficient representations. Unlike existing deep AL baselines [6, 27] that train deep models or even introduce additional discriminators [7, 28] in every data-selection round, our proposal only trains a light adapter, yielding simplified procedure with less computation and memory costs.

- As AL incorporates more labeled samples, we design a label-irrelevant patch augmentation scheme that preserves semantic information better than prior DAs. Together with our proposed framework, it gradually reduces redundant features and alleviates overfitting. Our DA scheme generally applies to different datasets and architectures and can potentially be extended to other learning tasks besides active learning.

- We extensively verified the improved performance on medical image classification tasks across various ViT architectures and AL strategies. Compared to existing widely-used AL paradigms, our proposed parameter-efficient AL framework can boost the overall performance of Few-shot AL by $5\% - 7\%$. Based on this framework, our proposed label-irrelevant patch augmentation methods can generally surpass existing DA methods by $1\% - 4\%$.

## 2. Related Work

**Self-supervised Learning (SSL).** SSL is used to derive feature representations from unlabeled samples [17, 29–31]. ExistingSSL tasks include predicting rotation angles [32], jigsaw puzzles [33], contrastive learning [34–36], and reconstruction-based training [16, 29, 30, 37, 38]. In our study, we broadly explored SSL and found the off-the-shelf ViTs to be superior (see Fig. 2), guiding our choice.

**Data Augmentation (DA).** DA is used to improve sample efficiency and mitigate overfitting [39, 40]. Traditional methods used simple transformations or augmentations altering the labels [41–43]. Recent methods learn to combine existing strategies or add consistency regularization [23, 44]. Most existing works concentrate on developing strategies to effectively combine these various transformations. AutoAug [23] uses a search algorithm to discover the optimal augmentation policies for the training dataset. RandAug [24] provides a simple and efficient strategy by randomly selecting

augmentation operations and magnitudes. However, none of them is designed to be label-irrelevant. Their potential alteration of the original semantic information has raised concerns [25, 45].

**Active Learning (AL).** The AL labeling strategies can be categorized into uncertainty-based and diversity-based methods. Uncertainty-based methods utilized the entropy [46], confidence [47], margin [48], and standard deviation [49] to measure informativeness. Diversity-based methods seek the most representative samples from the unlabeled dataset, such as Coreset [5] selected unlabeled samples that are furthest to their closest labeled samples, and Determinantal Point Process (DPP) [50] quantifies diversity based on a pairwise (dis)similarity matrix.

Prior works have used SSL for AL representation learning, including SSLAL [51], MoBYv2AL [52], and PT4AL [53]. These methods require SSL in each AL round, increasing time costs. We show that an off-the-shelf ViT already offers a sufficient representation, reducing the need for repeated SSL (Fig. 3). In integrating AL with DA, DAST-AL [54] and others [55–57] use various augmentation techniques. Yet, these often neglect potential semantic loss and aren't tailored for finetuning off-the-shelf ViTs that can outperform CNNs [58, 59].

# 3. Methodology

## 3.1. Setup

For an off-the-shelf ViT as $f_{\text{enc}}$ (off-the-shelf ViT pretrained as MAE also equips a decoder as $f_{\text{dec}}$), we denote the patchified input as $x \in \mathbb{R}^{U \times (P^2 \cdot C)}$, where $P$ and $U$ is the width and number of patches.

For AL, in $k$-th round, we denote the labeled/unlabeled set as $D_k^{\text{lab}}/D_k^{\text{unl}}$, respectively. The acquisition function $\alpha(x, \mathcal{M}_k)$ will output a value measuring the informativeness for $x$ in $D_k^{\text{lab}}$ according to the trained predictive model $\mathcal{M}_k$. Based on the outputted values, it can select a batch of $b$ samples in $D_k^{\text{unl}}$ as $B_k$, query their labels from human annotators and then move them from $D_k^{\text{unl}}$ to $D_k^{\text{lab}}$ with the queried labels. Such a process is illustrated as the blue lines in Fig. 4. As we introduced in section 2, existing AL strategies mainly focus on how to design acquisition function $\alpha(x, \mathcal{M}_k)$ to select the most informative samples. For example, the least confidence such as $\max_{\hat{y}} p_{\mathcal{M}_k}(\hat{y}|x)$ can be utilized to be the acquisition function, where $\hat{y}$ is the prediction with the highest probability.

## 3.2. Parameter-efficient AL on Off-the-shelf ViT

As illustrated in Fig. 4, in every AL round, we train an adapter $g$ on top of a pretrained and frozen ViT $f_{\text{enc}}$, where $g$ is designed to be a combination of a lightweight MLP $g_{\text{enc}}$ and a linear classification head $g_{\text{cls}}$. Within the $k$-th round, only $g$ is optimized via $D_k^{\text{lab}}$ while $f_{\text{enc}}$ is kept freezing, much more efficient than existing AL works that optimize the entire network [51–53] for each AL round. The AL acquisition function $\alpha(x, \mathcal{M}_k)$ is then conducted based on the trained predictive model as $\mathcal{M}_k(x) := g(f_{\text{enc}}(x))$. Here $g_{\text{enc}}$ is designed to gradually make the feature representations $\phi(x) := g_{\text{enc}}(f_{\text{enc}}(x))$ near minimally sufficient, and $g_{\text{cls}}$ aims to predict the labels based on $\phi(x)$.

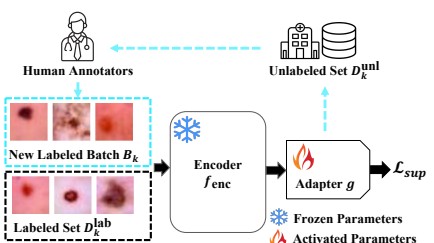

Figure 4: Parameter-efficient AL on off-the-shelf ViTs, where $\mathcal{L}_{\text{sup}}$ denotes the loss function for supervised training (e.g. cross-entropy)

## 3.3. Exploring the Diversity of Label-Irrelevant Patches

Based on our framework (Fig. 4), we further design a diverse set of data augmentations $A \in \mathcal{A}$ that are label-irrelevant. When $A$ is label irrelevant, it means $y|x \overset{d}{=} y|\phi(x) \overset{d}{=} y|\phi(\hat{x})$ (equivalent in distribution), where $\hat{x} := A(x)$ and $A$ is randomly sampled from $\mathcal{A}$. Therefore there is no bias when assigning the same label $y$ associated with $x$ to $\hat{x}$ and this ensures the representation $\phi$ to be sufficient. As we investigate a more diverse set of $\mathcal{A}$, the dependence between $\phi(x)$ and $\phi(\hat{x})$ is reduced, and

the more sample-efficient the learning procedure will be. To see why this helps, one can think about the most ideal case where $\phi(x) \perp \phi(\hat{x})|y$, meaning their dependence is only through $y$, and now the representation $\phi$ is minimal and it only requires to learn $y$ through $\phi(x)$ with order $|Y|$ samples, where $|Y|$ is the number of classes [12].

The diverse set of label-irrelevant data augmentations $\mathcal{A}$ enforces $\phi(x) := g_{\text{enc}}(f_{\text{enc}}(x))$ to be invariant to the redundant information in $x$, producing features closer to the minimal sufficient representation. Such representation nicely integrates with the AL procedure and the relationship between $x$ and $y$ will be learned rapidly and concretely. However, existing heuristic DA methods such as RandAug [24] do not satisfy label-irrelevance, causing the learned feature representation $\phi(x)$ not near minimally sufficient, which cannot learn the relation between $x$ and $y$ effectively during AL. We design a diverse set of label-irrelevant DA strategies by exploring the diversity of label-irrelevant patches. Given the patch-wise feature extraction approach taken by ViTs, enhancing the diversity of label-irrelevant patches allows a better approximation of minimally sufficient $\phi(x)$. Our proposed DA consists of two steps: **(1) Localize the label-irrelevant patches**, and **(2) Augment the label-irrelevant patches**. The details of **(1)** and **(2)** will be introduced as follows.

### 3.3.1. Label-Irrelevant Patches Localization

We localize the label-irrelevant patches by computing a patch-wise correlation vector $\text{Cor} \in \mathbb{R}^U$ between $x$ and $y$, where its $i$-th coordinate $\text{Cor}_i$ indicates the correlation between the $i$-th patch and $y$. Existing model explanation techniques such as DeepLIFT [60], Saliency [61], CosineAttention-Map [62], and LastAttentionMap [63], can generate a saliency map that illuminates the semantic regions of a raw image $x$ regarding ground truth label $y$. Hence, we compute Cor by segmenting the saliency map into patches, aggregating values at the patch level, and subsequently normalizing the result. To establish a criterion for identifying irrelevant patches of $x$, we operationally select them by masking the lowest $r\%$ (e.g. 75%) patches by $M_{\text{Cor}}$ as $\bar{x} = M_{\text{Cor}} \odot x$, where $M_{\text{Cor}}$ is a patch-wise 0-1 mask generated to mask the lowest $r$ patches in $x$ in regard to Cor. More discussions for these localization methods are deferred to section 4.2 and appendix A.3.

### 3.3.2. Label-Irrelevant Patches Augmentation

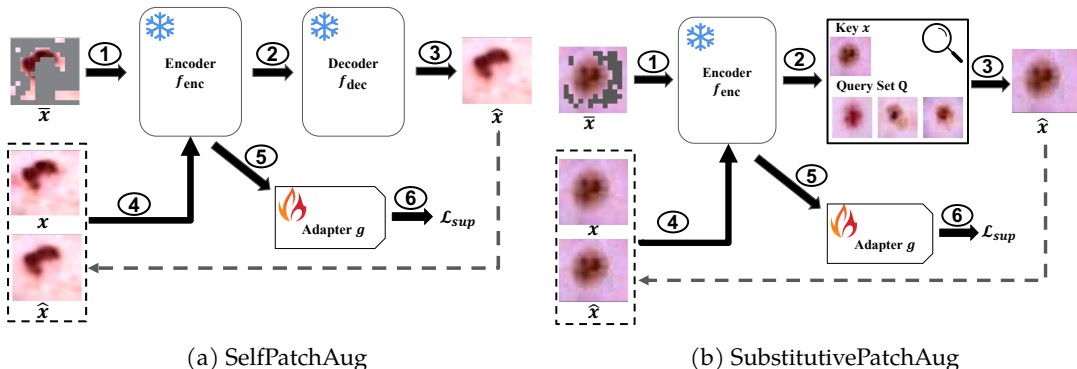

(a) SelfPatchAug           (b) SubstitutivePatchAug

Figure 5: We propose two different augmentation methods to explore the diversity of label-irrelevant patches as **(a) SelfPatchAug** and **(b) SubstitutivePatchAug**. ① ④ ⑤ ⑥ indicate same operations in both **(a)** and **(b)**, where ① is the forward process, and ④ ⑤ ⑥ represent the training of adapter $g$. ② ③ are the core operations for **(a)** and **(b)**, respectively.

For label-irrelevant DA as $\hat{x} := A(x)$, we primarily emphasize two effective methods (**SelfPatchAug** and **SubstitutivePatchAug**) in this section to augment the label-irrelevant patches localized by $M_{\text{Cor}}$. Both of these two DA methods satisfy label-irrelevance since they only explore the diversity of label-irrelevant patches while keeping label-relevant patches unchanged. Moreover, we present **Instance-adaptive Label Smoothing (IaLS)** alongside these two label-irrelevant DA methods to achieve better performance by alleviating the negative impact caused by patch augmentation.

**SelfPatchAug.** For SefPatchAug (Fig. 5a), we augment the localized label-irrelevant patches by reconstructing them via an encoder-decoder structure. As shown in Fig. 5a ①, ② and ③, the patchified augmented sample $\hat{x}$ can be produced as $\hat{x} = A(x) = \bar{x} + f_{\text{dec}}(f_{\text{enc}}((1 - M_{\text{Cor}}) \odot x)))$. In each AL round, training the adapter $g$ on the augmented features $f_{\text{enc}}(\hat{x})$ enables the representations $\phi(x) := g_{\text{enc}}(f_{\text{enc}}(x))$ to be more minimally sufficient, leading to a classifier $g_{\text{cls}}$ with better generalizability.

However, the limitation of SelfPatchAug is that it can only be applied to the off-the-shelf ViTs with checkpoints released by MAE due to the requirement of pretrained decoder $f_{\text{dec}}$. Differently, SubstitutivePatchAug is compatible with off-the-shelf ViTs pretrained with various SSL strategies (e.g. DINO, MoCo-V3), which will be introduced as follows.

**SubstitutivePatchAug.** SubstitutivePatchAug is inspired by existing DA methods for text data [64, 65]. In the field of text classification, identifying suitable word substitutions and replacing the original words with substitutions is a kind of prevalent DA strategy [66–68]. The structure of Transformers is originally designed for NLP tasks [69], and ViTs treat an image as a sequence of non-overlapping patches, just like how Transformers handle tokens in a sentence. Therefore, the intuition of SubstitutivePatchAug is that we take the scope of ViTs and consider patches as words. Thus this method augments the label-irrelevant patches by substituting them with semantically related patches from $t$ similar images from a query set $Q$ selected from $D = D_k^{\text{unl}} \cup D_k^{\text{lab}}$.

As shown in Fig. 5b ②, we compute a pre-defined similarity matrix $\Psi \in \mathbb{R}^{N \times N}$ among $D$, where $\Psi(i, j)$ indicates the similarity between $f_{\text{enc}}(x_i)$ and $f_{\text{enc}}(x_j)$, and $N$ is the number of samples in $D$. With $x$ as a key, we search the top-$t$ samples as a query set $Q = \{(x_i, y_i)\}_{i=1}^t$ according to $\Psi$. In Fig. 5b ③, by leveraging query set $Q$ and key $x$, we construct a feature-level/raw-level patch similarity matrix $\Phi_{\text{rep}}/\Phi_{\text{raw}} \in \mathbb{R}^{U \times (t \times U)}$ based on $f_{\text{enc}}(x)/x$, referring to the substitutive patches defined from the features space and raw-data space, respectively. Subsequently, we use a trade-off factor $\lambda \in [0, 1]$ for linear combination as $\Phi = \lambda \Phi_{\text{raw}} + (1 - \lambda)\Phi_{\text{rep}}$. Such linear combination aims to select substitutive patches that can balance between the similarities in feature space and raw-data space. For every patch in $x$, we can select the most similar patch from $Q$ by $\Phi$, hence produce the augmented image $\hat{x} = A(x)$ by filling in the masked patches of $\bar{x}$ with the selected substitutive patches.

**Instance-adaptive Label Smoothing (IaLS) for Augmented Data** Inspired by existing works [70, 71] that utilize label smoothing [72] to alleviate the degradation of semantic information caused by DA, we propose an instance-adaptive label smoothing (IaLS) strategy for the augmented image $\hat{x}$. The primary motivation behind IaLS is that, since label-irrelevant patches are selected by Cor with a hard threshold ratio of $r\%$, reconstructing (SelfPatchAug) or replacing (SubstitutivePatchAug) these patches risks of losing some label-relevant information. In other words, $\phi(x) \perp \phi(\hat{x})|y$ is merely an idealized condition, and cannot be guaranteed with absolute certainty in practical applications, particularly when $(1 - M_{\text{Cor}})^T \text{Cor}$ is large.

To address this issue, we introduce an instance-adaptive factor $\beta_x$ to reduce the confidence of the model's prediction for the augmented image $\hat{x}$. Specifically, the smoothed label vector of the augmented sample $\hat{x}$ is computed as $\hat{y} = \beta_x \cdot \frac{1}{|Y|} + (1 - \beta_x) \cdot y$. For each image $x$ with its corresponding $M_{\text{Cor}}$ and Cor, we smooth its ground truth vector by an instance-adaptive factor $\beta_x = (1 - M_{\text{Cor}})^T \text{Cor}$.

# 4. Experiments

## 4.1. Experimental Setups

**Datasets.** We evaluate our methods on MedMNIST [73], an ensembled evaluation benchmark encompassing various classification tasks for medical imaging. MedMNIST is a widely adopted collection of standardized biomedical image datasets designed for image classification tasks. We conduct our methods on DermaMNIST, BloodMNIST, PneumoniaMNIST, OrganAMNIST, OrganCMNIST, and OrganSMNIST, respectively.

**AL Settings.** We conduct our experiments with various AL strategies to demonstrate that our method can achieve promising performance consistently across different AL strategies. Our work investigates

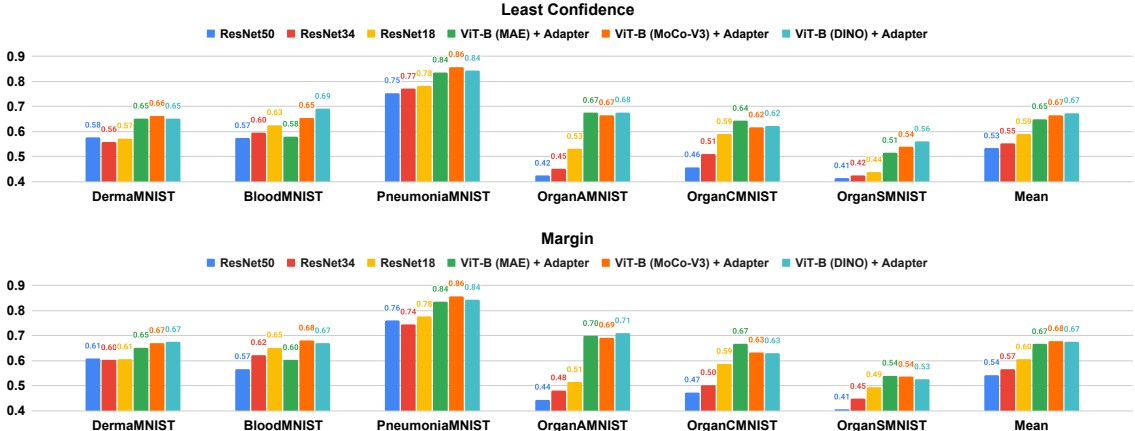

Figure 6: The AUBC results of comparing existing AL model structures (ResNet) and our efficient model structures (ViT-B + Adapter) via AL strategies as *Least Confidence* and *Margin*. **Mean** here denotes the averaged values of results among 6 different datasets.

| | DermaMNIST | BloodMNIST | PneumoniaMNIST | OrganAMNIST | OrganCMNIST | OrganSMNIST | Mean | Avg Rank |
|---|---|---|---|---|---|---|---|---|
| | | | Least Confidence | | | | | |
| ViT-B (MAE) | 0.6513 | 0.5790 | 0.8350 | 0.6740 | 0.6420 | 0.5140 | 0.6492 | 4.17 |
| + RandAug | 0.6400 | 0.5800 | **0.8410** | 0.6840 | **0.6760** | 0.5290 | 0.6583 | 3.00 |
| + AutoAug | 0.6410 | 0.6090 | 0.8160 | **0.7050** | 0.6410 | 0.5120 | 0.6540 | 3.83 |
| + NormalAug | 0.6330 | 0.6220 | 0.8400 | 0.5440 | 0.5410 | 0.5110 | 0.6152 | 4.83 |
| + SubstitutivePatchAug | 0.6700 | **0.6340** | 0.8280 | 0.6850 | 0.6710 | **0.5360** | **0.6707** | 2.33 |
| + SelfPatchAug | **0.6920** | 0.6270 | 0.8330 | 0.6840 | 0.6390 | 0.5170 | 0.6653 | 2.67 |
| ViT-B (MoCo-V3) | 0.6620 | 0.6540 | 0.8570 | 0.6650 | 0.6160 | 0.5380 | 0.6653 | 3.83 |
| + RandAug | 0.6420 | **0.7020** | **0.8610** | 0.6790 | 0.6380 | 0.5430 | 0.6775 | 2.67 |
| + AutoAug | 0.6690 | 0.6960 | 0.8490 | 0.6870 | 0.6140 | 0.5510 | 0.6777 | 2.83 |
| + NormalAug | **0.6720** | 0.6750 | 0.8220 | 0.5560 | 0.5740 | 0.5460 | 0.6408 | 3.50 |
| + SubstitutivePatchAug | 0.6640 | 0.6890 | 0.8270 | **0.6930** | **0.6480** | **0.5830** | **0.6840** | 2.17 |
| ViT-B (DINO) | 0.6510 | 0.6900 | 0.8420 | 0.6760 | 0.6220 | 0.5590 | 0.6733 | 3.83 |
| + RandAug | 0.6590 | 0.6920 | 0.8670 | 0.6980 | 0.6490 | 0.5340 | 0.6832 | 2.67 |
| + AutoAug | 0.6440 | 0.6920 | **0.8700** | 0.6900 | 0.6490 | 0.5510 | 0.6827 | 3.17 |
| + NormalAug | 0.6740 | **0.6980** | 0.8250 | 0.5670 | 0.5540 | 0.5260 | 0.6407 | 3.83 |
| + SubstitutivePatchAug | **0.6760** | 0.6940 | 0.8600 | **0.7020** | **0.6750** | **0.5770** | **0.6973** | **1.50** |
| | | | Margin | | | | | |
| ViT-B (MAE) | 0.6517 | 0.6020 | 0.8350 | 0.6980 | 0.6670 | 0.5400 | 0.6656 | 4.50 |
| + RandAug | 0.6737 | 0.6240 | **0.8410** | 0.7200 | 0.6980 | 0.5500 | 0.6845 | 2.50 |
| + AutoAug | 0.6463 | 0.6170 | 0.8160 | 0.7190 | 0.6690 | 0.5400 | 0.6679 | 4.83 |
| + NormalAug | 0.6427 | 0.6230 | 0.8400 | 0.6427 | 0.5940 | 0.5270 | 0.6449 | 5.00 |
| + SubstitutivePatchAug | 0.6640 | 0.6410 | 0.8280 | 0.7240 | **0.7270** | **0.5850** | **0.6948** | 2.17 |
| + SelfPatchAug | **0.6880** | **0.6470** | 0.8330 | 0.7220 | 0.7000 | 0.5650 | 0.6925 | **2.00** |
| ViT-B (MoCo-V3) | 0.6700 | 0.6800 | **0.8570** | 0.6920 | 0.6320 | 0.5350 | 0.6777 | 3.50 |
| + RandAug | 0.6610 | 0.7100 | 0.8560 | 0.7280 | 0.6670 | 0.5700 | 0.6987 | 2.50 |
| + AutoAug | 0.6570 | 0.7150 | 0.8490 | 0.7160 | 0.6610 | 0.5590 | 0.6928 | 3.17 |
| + NormalAug | 0.6660 | 0.6970 | 0.8320 | 0.5500 | 0.5910 | 0.5500 | 0.6477 | 4.17 |
| + SubstitutivePatchAug | **0.6940** | **0.7230** | 0.8270 | **0.7500** | **0.6880** | **0.5880** | **0.7117** | **1.67** |
| ViT-B (DINO) | 0.6740 | 0.6690 | 0.8420 | 0.7090 | 0.6290 | 0.5250 | 0.6747 | 4.33 |
| + RandAug | 0.6610 | 0.7110 | 0.8470 | 0.7220 | 0.6670 | 0.5450 | 0.6922 | 3.33 |
| + AutoAug | 0.6760 | 0.7070 | **0.8690** | 0.7280 | 0.6680 | 0.5640 | 0.7020 | 2.33 |
| + NormalAug | 0.6770 | 0.7090 | 0.8250 | 0.5740 | 0.5780 | 0.5560 | 0.6532 | 3.83 |
| + SubstitutivePatchAug | **0.6810** | **0.7200** | 0.8560 | **0.7610** | **0.7040** | **0.6050** | **0.7212** | **1.17** |

Table 1: Results of the comparison between our proposed label-irrelevant patch augmentation methods and other DA methods across 6 datasets for medical image classification by utilizing *Least Confidence* and *Margin* as the AL strategy. We conduct those DA methods on our efficient AL framework via different off-the-shelf ViTs. Note that **Mean** represents the averaged AUBC across 6 datasets, while **Avg Rank** is computed by ranking the AUBC performance on each dataset and then taking the average.

two AL paradigms: Few-shot AL ($N_0^{lab} = 10, K = 50, b = 5$) and Many-shot AL ($N_0^{lab} = 1000, K = 5, b = 500$). Notably, Few-shot AL poses a sterner challenge due to its greater potential of overfitting, induced by the paucity of labeled data especially at the early stage of AL. Besides, Few-shot AL is more practical in performing medical image classification due to the high labeling cost. Due to the page limitation, all the results posted in tables and figures in section 4.2 are produced under the paradigm of Few-shot AL, and the results of Many-shot AL will be presented in the Appendix.

**Evaluation Metrics.** To evaluate the performance of AL, we report *area under the budget curve* (*AUBC*) [74] in our experimental results, where the AUBC value is calculated by the trapezoid

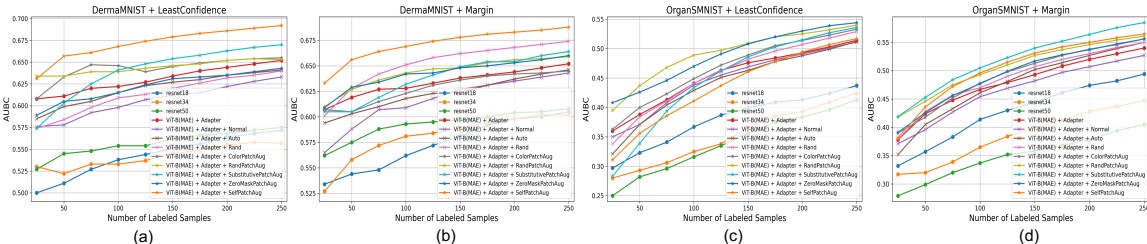

Figure 7: The AUBC curve for DermaMNIST and OrganSMNIST based on the AL strategies of *Least Confidence* and *Margin*. The lines marked with **'-o-'** denote the widely used baselines in AL, and the lines marked with **'-x-'** represent existing DA methods while the lines with **'-\*-'** indicate various label-irrelevant patch augmentation methods based on our efficient AL framework.

method over a given budget curve, and higher AUBC values indicate better overall performances for varying budgets during the AL process.

## 4.2. Experimental Results and Analysis

**Effectiveness and Efficiency of Our AL Framework.** As Fig. 6 illustrated, to show the superiority of our proposed efficient AL framework (Fig. 4), we employ ResNet18/34/50 as the baseline models for comparison across various AL strategies. We choose ResNet18/34/50 as

|  | ResNet18 | ResNet34 | ResNet50 | Adapter (Ours) |
|---|---|---|---|---|
| Parameters Size ($10^9$) | 1.8240 | 3.6787 | 4.1337 | **0.0006** |

Table 2: Comparison of parameters size between existing AL structures (ResNet) and our proposed framework (Adapter).

baselines due to their varying degrees of parameter complexity and widespread use in the AL field. Results illustrated in Fig. 6 show that our proposed efficient AL framework on off-the-shelf ViTs can surpass the selected baselines via various AL methods across six datasets selected from MedMNIST. Comparing the results of off-the-shelf ViTs (MAE/MoCo-V3/DINO) between Fig. 6 and Fig. 2, it obviously supports our claim in section 1 that sufficient feature representations are the key to a successful AL, since the relative magnitude of AUBC shown in Fig. 6 is almost consistent with the relative magnitude of LR/KNN Accuracy shown in Fig. 2, demonstrating that sufficient representations lead the AL acquisition function to select promising informative samples. Besides, in Table 2, we provide a comparative analysis of the parameter size for several widely-used AL architectures, where our framework demonstrates a significant reduction in the number of parameters.

**Effectiveness of SelfPatchAug & SubstitutivePatchAug.** Based on our proposed efficient framework (Fig. 4), we further compare the performance of our label-irrelevant patch augmentation methods against existing data augmentation techniques, such as AutoAug and RandAug. Additionally, we design a practical DA strategy as a baseline named NormalAug by simply combining random horizontal and vertical flips.

Our label-irrelevant patch augmentations (SubstitutivePatchAug and SelfPatchAug), as shown in Table 1, markedly outperform existing DA methods in Few-shot AL across various medical datasets. This success, reflected in the **Mean** and **Avg Rank** metrics, underscores the significant and consistent enhancement our methods bring to our efficient AL framework. The Mean metric together with the Avg Rank offers a comprehensive evaluation of the DA methods. While our SubstitutivePatchAug and SelfPatchAug methods excel in most datasets, they falter in cases like PneumoniaMNIST where lesions pervade the entire image, making almost all patches label-relevant and risking the loss of semantic information through any patch augmentation. However, in datasets like DermaMNIST, where lesions are localized in some specific regions, our methods demonstrate effectiveness by safely augmenting label-irrelevant patches.

**Flexibility of Label-irrelevant Patch Augmentation.** Besides the label-irrelevant DA methods we mentioned in section 3.3.2, our label-irrelevant patch augmentation approach exhibits high flexibility for various DA methods. We can plug some other DA methods including color jitter (ColorPatchAug),

| ViT-B (MAE) | Least Confidence | | | | Margin | | | |
|---|---|---|---|---|---|---|---|---|
| | LastAtt | CosineAtt | Saliency | DeepLIFT | LastAtt | CosineAtt | Saliency | DeepLIFT |
| $r = 25\%$ | 0.675 | **0.678** | 0.648 | 0.668 | 0.640 | 0.653 | 0.646 | 0.613 |
| $r = 50\%$ | 0.655 | 0.665 | 0.654 | 0.657 | 0.687 | 0.671 | 0.663 | 0.658 |
| $r = 75\%$ | 0.643 | 0.670 | 0.675 | 0.638 | 0.672 | 0.664 | 0.662 | **0.681** |

Table 3: AUBC results for SubstuitiveAug via ViT-B (MAE) for different $r$ and label-irrelevant patch localization methods on DermaMNIST. The values marked as blue are the results presented in Table 1

.

RandAug (RandPatchAug), and zero-masking (ZeroMaskPatchAug) to explore the diversity of the localized label-irrelevant patches. These methods are called as **extended label-irrelevant patch augmentation methods** as follows.

Fig. 7 illustrates the curve for AUBC results for varying budgets during the AL process, where extended patch augmentation methods perform competitively. In most cases like Fig. 7(a)(b)(d), most extended patch augmentation methods perform better AUBC results on varying budgets than existing DA methods (RandAug/AutoAug/NormalAug), but underperform SubstitutivePatchAug and SelfPatchAug. However, as shown in Fig. 7(c), some of the extended patch augmentation methods like RandPatchAug and ZeroMaskPatchAug can even surpass our carefully designed label-irrelevant patch augmentation methods (SelfPatchAug & SubssituitivePatchAug), showing the great potential of the way to plug different DA methods in the flexible label-irrelevant patch augmentation approach.

**Do We Need Pretext SSL Training for the Downstream Task?** As we claimed in section 1 and section 2, one key difference between our work and existing AL+SSL works is that our method does not require SSL pretext training on the downstream dataset $D$.

| | Least Confidence | Margin | Least Confidence MC | Coreset |
|---|---|---|---|---|
| ViT-B (MAE)-Origin | **0.6513** | **0.6517** | **0.6653** | **0.6387** |
| ViT-B (MAE)-Adapted | 0.6147 | 0.616 | 0.6153 | 0.607 |

Table 4: AUBC results on DermaMNIST with different AL strategies for **ViT-B (MAE)-Origin** and **ViT-B (MAE)-Adapted**.

The reason is that conducting pretext SSL training on the medical image dataset is not efficient (costing time for training) and can even distort the representations (Fig. 3). This statement was further demonstrated by Table 4, where ViT-B (MAE)-Adapted performed worse than ViT-B (MAE)-Origin with respect to AUBC for various AL strategies on DermaMNIST.

**Different $r$ and Localization Methods.** The ablation study for different $r$ and localization methods are shown in Table 3. We report the results marked as blue (not the best) in Table 1 since we need to maintain consistent hyperparameters across datasets for fair comparison. It's important to note that dataset-specific hyperparameter tuning could further enhance the performance of the label-irrelevant patch augmentation. As shown in Table 3, in most cases, CosineAttentionMap and LastAttentionMap can outperform DeepLIFT and Saliency with respect to the AUBC results. However, for *Margin* sampling with $r = 75\%$, DeepLIFT performs much better than others.

In terms of efficiency, CosineAttentionMap and LastAttentionMap outperform DeepLIFT and Saliency for localizing label-irrelevant patches, as they are produced simultaneously during the forward process without additional computational costs, while DeepLIFT and Saliency require multiple backward propagations, significantly increasing time cost.

# 5. Conclusion

In this paper, we argue that the key to successful AL is to learn a minimally sufficient representation. We presented an efficient AL framework leveraging off-the-shelf ViTs to gain a relatively good representation at the initial stage of AL. We further propose a DA method for localizing and augmenting label-irrelevant patches, to gradually train a lightweight encoder to transfer the representation closer to minimally sufficient. The effectiveness and efficiency of our framework are widely evaluated across various datasets and AL strategies.

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

# A. Details

In the appendix, we present the details of datasets, hyperparameters, metrics, and our algorithm in appendix A.1 and appendix A.2. Details of the label-irrelevant patch location methods can be found in appendix A.3, and other patch augmentation methods besides SelfPatchAug and SubstitutivePatchAug can be found in appendix A.4. For additional experimental results, appendix B.1 contains results for more comparison and ablation studies, and appendix B.2 includes the results for visualization studies.

## A.1. Details of Experimental Settings

**Details of Datasets.** The details of the datasets we used in our experiments are presented in Table 5. These datasets comprise medical imaging data from diverse modalities, varying numbers of classes, distinct pathological conditions, and different anatomical regions, providing a comprehensive evaluation of our proposed framework.

| | Data Modality | Number of Class | Training / Test |
|---|---|---|---|
| DermaMNIST | Dermatoscope | 7 | 7007 / 2005 |
| BloodMNIST | Blood Cell Microscope | 8 | 11959 / 3421 |
| PneumoniaMNIST | Chest X-Ray | 2 | 4708 / 624 |
| OrganAMNIST | Abdominal CT | 11 | 34581 / 17778 |
| OrganCMNIST | Abdominal CT | 11 | 13000 / 8268 |
| OrganSMNIST | Abdominal CT | 11 | 13940 / 8829 |

Table 5: Details of Datasets

**Details of Hyperprameter Settings.** For Tables 1 and 4, and Figs. 6 and 7. The results of SelfPatchAug are produced by fixed $r = 75\%$, and the results of SubstitutivePatchAug are produced by $r = 75\%$, $t = 5$, and $\lambda = 0.5$.

**Details of Computing Resources.** All the experiments can be run on a single NVIDIA A100 (80GB)

**Details of Metrics.** In this section, we will introduce the metrics evaluating the representation quality presented in Fig. 2 and Fig. 3.

- **LR Accuracy.** This metric is utilized to verify the linear separability of the learned feature representation space by modeling $P(y|\phi(x))$ with LR. For the implementation details of the LR classifier, we use *LogisticRegression*(*random_state=66,solver='sag'*) from *sklearn*.

- **KNN Accuracy.** Similar to LR Accuracy, it measures the representation quality by modeling $P(y|\phi(x))$ with KNN. We use the *KNeighborsClassifier*(*n_neighbors=5,algorithm='auto'*) from *sklearn* as the KNN classifier.

- **Representation Metric.** We denote a matrix as $M \in \mathbb{R}^{c \times c}$, where $c$ is the number of classes. For $M_{i,j}$, it represents a score measuring the distance between the representations from $i - th$ class and $j - th$ class center, which is computed as eq. (1)

$$M_{i,j} = \frac{1}{N_i} \sum_{z \in \{z|y=i\}} S(z; u_j), \quad u_j = \frac{1}{N_j} \sum_{z \in \{z|y=j\}} z \tag{1}$$

where $S(a; b)$ denotes the Euclidean distance between vector $a$ and $b$. Generally, we compute the intra-class and inter-class distance based on $M$, and then the representation metric $\tau$ shown in Fig. 3 is defined as eq. (2).

$$\tau_{intra} = \frac{Trace(M)}{C}, \quad \tau_{inter} = \frac{Sum(M) - Trace(M)}{C}, \quad \tau = \frac{\tau_{inter}}{\tau_{intra} + \tau_{inter}} \tag{2}$$

where a larger $\tau$ indicates a better feature representation space.

## A.2. Algorithm of Our Framework

---

**Algorithm 1** AL with Label-irrelevant Patches Augmentation

---

**Input:** Initial Labeled/Unlabeled Dataset $D_0^{\text{lab}}/D_0^{\text{unl}}$, AL Rounds $K$, Off-the-shelf Encoder/Decoder $f_{\text{enc}}/f_{\text{dec}}$, Adapter $g$,
1: $r \leftarrow a \mod b$
2: **for** $k = 0$ to $K$ **do**
3:    **for** $\{x, y\} \in D_k^{\text{lab}}$ **do**
4:       $\text{Cor} = \text{Localize}(x, y; f_{\text{enc}}, f_{\text{dec}}, g)$
5:       $\bar{x}_p = M_{\text{Cor}} \odot x_p$    */\*Localize the Label-irrelevant Patches\*/*
6:       $\hat{x} = A(x)$    */\*Argument the Label-irrelevant Patches\*/*
7:       $\beta_x = (1 - M_{\text{Cor}})^T \text{Cor}$
8:       $\hat{y} = \beta_x \cdot \frac{1}{|Y|} + (1 - \beta_x) \cdot y$    */\*Instance-adaptive Label Smoothing\*/*
9:       $\hat{D}_k^{\text{lab}} = D_k^{\text{lab}} \cup \{\hat{x}, \hat{y}\}$
10:   **end for**
11:   Train $g$ on $\hat{D}_k^{\text{lab}}$
12:   $B = \alpha(D_k^{\text{unl}}, [f_{\text{enc}}, f_{\text{dec}}, g])$
13:   $D_{k+1}^{\text{lab}} = D_k^{\text{lab}} \cup B$    */\*Acquire Unlabeled Samples\*/*
14:   $D_{k+1}^{\text{unl}} = D_{k+1}^{\text{unl}} \setminus B$
15: **end for**
**Output:** Adapter $g$

---

The overall algorithm of our efficient AL framework with label-irrelevant augmentation is shown in algorithm 1, where $\alpha(D_k^{\text{unl}}, [f_{\text{enc}}, f_{\text{dec}}, g])$ denotes selecting batch $B$ from $D_k^{\text{unl}}$ based on the acquisition function $\alpha$.

## A.3. Details of Label-Irrelevant Patches Localization Methods

- **LastAttentionMap:** This method generates attention maps using the last layer of the network. These maps are useful in visualizing and understanding where the network is focusing its attention while making predictions.

- **CosineAttentionMap:** This technique generates attention maps using cosine similarity, which measures the cosine of the angle between two vectors. It can highlight the most influential regions in the input for the output prediction.

- **Saliency:** Saliency maps are a common approach in visualizing and interpreting neural networks. These maps show the gradient of the output with respect to the input image, giving an indication of which pixels contribute most to the network's decision.

- **DeepLIFT:** DeepLIFT is a method for computing the contributions of inputs to outputs, given a neural network. It assigns contribution scores by comparing the activation of each neuron to its activations and computing the differences. This helps to identify which parts of the input are important for prediction.

The results of the ablation study for these four different localization methods are presented in the Table 3.

## A.4. Label-Irrelevant Patches Augmentation Methods

Many augmentation methods have been used for image classification tasks. We compared our augmentation methods with the following state-of-the-art methods:

- **AutoAug**: AutoAug first adopts a search phase that uses reinforcement learning to do the choose best operations of augmentation.

- **RandAug**: RandAug reduces the complexity of the augmentation process by removing the research phase, which reduces the parameter space to a fixed uniform possibility for every operation of transformation and a universal magnitude parameter.
- **NormalAug**: NormalAug applies a combination of random horizontal and vertical flips to the images.

We also plugged some other DA methods, which we call extended patch augmentation methods:

- **ColorPatchAug**: Color jitter is applied to label irrelevant patches.
- **RandPatchAug**: RandAug is applied to the label irrelevant patches.
- **ZeroMaskPatchAug**: Mask the label irrelevant patches with zeros.

## A.5. Details of AL Strategies

We conducted our experiments with different acquisition functions $\alpha(x, \mathcal{M}_k)$ utilized as the query strategies in active learning as follows:

- **LeastConfidence**: Least Confidence method chooses samples whose predicted labels the current model is least certain with. The acquisition function for Least Confidence method can be denoted as $\alpha_{\text{LeastConfidence}}(x, \mathcal{M}_k) = -\max_{\hat{y}} p_{\mathcal{M}_k}(\hat{y}|x)$, where $\hat{y}$ is the prediction with the highest probability.
- **Margin**: Margin Sampling Looks into the first and second most likely predicted labels of unlabeled samples, and selects those where the difference in probability between the top two predicted labels is relatively small. The acquisition function for Margin Sampling can be denoted as $\alpha_{\text{Margin}}(x, \mathcal{M}_k) = -(p_{\mathcal{M}_k}(\hat{y}_1|x) - p_{\mathcal{M}_k}(\hat{y}_2|x))$, where $\hat{y}_1, \hat{y}_2$ is the two most likely labels of $x$.
- **Entropy**: Maximum Entropy Sampling selects samples with the most significant entropy loss. The acquisition function for Maximum Entropy Sampling can be denoted as $\alpha_{\text{MaximumEntropy}}(x, \mathcal{M}_k) = -\sum_c p_{\mathcal{M}_k}(y = c|x)\log p_{\mathcal{M}_k}(y = c|x)$, where $c$ denotes the classes.
- **CoreSet**: CorrSet methods select the most representative samples by selecting a set of center points and minimizing the distance from any point in the dataset to its closest center point. This is equivalent to minimizing the difference between the average loss calculated over the selected center points and the average loss calculated over the entire dataset. The acquisition function for Coreset can be denoted as $\alpha_{\text{Coreset}}(x, \mathcal{M}_k) = \max_{x_i \in D_k^{\text{lab}}} d(\mathcal{M}_k(x), \mathcal{M}_k(x_j))$, where $d(\cdot, \cdot)$ is distance metric and $\mathcal{M}_k(x)$ denotes the representation of $x$ encoded by $\mathcal{M}_k$.

# B. Additional Experimental Results

## B.1. More Ablation Studies

**Results for Many-shot AL.** The results of Many-shot AL are shown in Table 6 and Fig. 8. Compared to Few-shot AL, our DA method delivers more modest enhancements, but it continues to outshine existing DA techniques.

**More Results for Few-shot AL.** More results for different AL strategies under the setting of Few-shot AL are presented in Table 7 and Fig. 9. It shows the effectiveness of our method for boosting the performances for various AL strategies consistently.

**Effectiveness of Instance-adaptive Label Smoothing.** The ablation study of Instance-adaptive Label Smoothing (IaLS) are shown in Table 8 conducted on ViT-B (MAE), which demonstrate the effectiveness of IaLS.

**Structure of Adapter $g$.** We also explore different structures for the adapter $g$ shown in Table 9, where the **ResAdapter** is designed as adding a skip connection between $g_{\text{enc}}$ and $g_{\text{cls}}$.

| | DermaMNIST | PneumoniaMNIST | OrganAMNIST | OrganCMNIST | OrganSMNIST | **Mean** |
|---|---|---|---|---|---|---|
| | | | Least Confidence | | | |
| ViT-B (MAE) | 0.7353 | 0.8460 | 0.8820 | 0.8757 | 0.7230 | 0.8124 |
| + RandAug | **0.7450** | **0.8810** | 0.9020 | **0.8903** | 0.7350 | **0.8307** |
| + AutoAug | 0.7367 | 0.8600 | **0.8960** | 0.8860 | 0.7400 | 0.8237 |
| + NormalAug | 0.7397 | 0.8440 | 0.7730 | 0.7887 | 0.7190 | 0.7729 |
| **+ SubstitutivePatchAug** | 0.7447 | 0.8530 | **0.9050** | **0.8940** | **0.7530** | 0.8299 |
| **+ SelfPatchAug** | **0.7420** | 0.8480 | 0.8890 | 0.8910 | 0.7400 | 0.8220 |
| ViT-B (MoCo-V3) | 0.7310 | 0.8600 | 0.8970 | 0.8530 | 0.7340 | 0.8150 |
| + RandAug | 0.7380 | **0.8830** | 0.8930 | 0.8680 | 0.7530 | 0.8270 |
| + AutoAug | 0.7430 | 0.8760 | 0.8970 | 0.8660 | 0.7500 | 0.8264 |
| + NormalAug | **0.7410** | 0.8710 | 0.7530 | 0.7660 | 0.7370 | 0.7736 |
| **+ SubstitutivePatchAug** | 0.7430 | 0.8670 | **0.9070** | **0.8800** | **0.7610** | **0.8316** |
| ViT-B (DINO) | 0.7470 | 0.8770 | 0.8920 | 0.8630 | 0.7370 | 0.8232 |
| + RandAug | 0.7530 | **0.8820** | 0.9040 | 0.8780 | 0.7560 | 0.8346 |
| + AutoAug | 0.7510 | 0.8810 | 0.9060 | 0.8830 | 0.7510 | 0.8344 |
| + NormalAug | 0.7500 | 0.8620 | 0.7530 | 0.7730 | 0.7270 | 0.7730 |
| **+ SubstitutivePatchAug** | **0.7620** | 0.8680 | **0.9160** | **0.8870** | **0.7590** | **0.8384** |
| | | | Margin | | | |
| ViT-B (MAE) | 0.7373 | 0.8390 | 0.8860 | 0.8770 | 0.7240 | 0.8127 |
| + RandAug | 0.7433 | **0.8770** | 0.9020 | 0.8917 | 0.7450 | **0.8318** |
| + AutoAug | 0.7417 | 0.8670 | 0.8960 | 0.8910 | 0.7410 | 0.8273 |
| + NormalAug | 0.7427 | 0.8420 | 0.7670 | 0.7893 | 0.7190 | 0.7720 |
| **+ SubstitutivePatchAug** | 0.7433 | 0.8620 | **0.9080** | **0.8960** | **0.7490** | 0.8317 |
| **+ SelfPatchAug** | **0.7440** | 0.8510 | 0.8950 | 0.8850 | 0.7340 | 0.8218 |
| ViT-B (MoCo-V3) | 0.7340 | 0.8600 | 0.8860 | 0.8550 | 0.7300 | 0.8130 |
| + RandAug | **0.7410** | 0.8750 | 0.8930 | 0.8640 | 0.7520 | 0.8250 |
| + AutoAug | 0.7200 | **0.8770** | 0.8950 | 0.8710 | 0.7500 | 0.8226 |
| + NormalAug | 0.7350 | 0.8730 | 0.7500 | 0.7620 | 0.7440 | 0.7728 |
| **+ SubstitutivePatchAug** | **0.7410** | 0.8670 | **0.8960** | **0.8850** | **0.7580** | **0.8294** |
| ViT-B (DINO) | 0.7380 | 0.8770 | 0.8940 | 0.8670 | 0.7340 | 0.8220 |
| + RandAug | 0.7490 | **0.8810** | 0.9090 | 0.8790 | 0.7520 | 0.8340 |
| + AutoAug | 0.7510 | 0.8800 | 0.9060 | 0.8790 | 0.7510 | 0.8334 |
| + NormalAug | 0.7500 | 0.8650 | 0.7550 | 0.7690 | 0.7300 | 0.7738 |
| **+ SubstitutivePatchAug** | **0.7550** | 0.8700 | **0.9160** | **0.8880** | **0.7590** | **0.8376** |

Table 6: Results of the comparison between our proposed label-irrelevant patch augmentation methods and other DA methods across 6 datasets for medical image classification by utilizing *Least Confidence* and *Margin* as the AL strategy, under the setting of Many-shot AL.

## B.2. Visualization Studies

The visualization results of our proposed label-irrelevant DA methods are shown in Fig. 10 and Fig. 11.

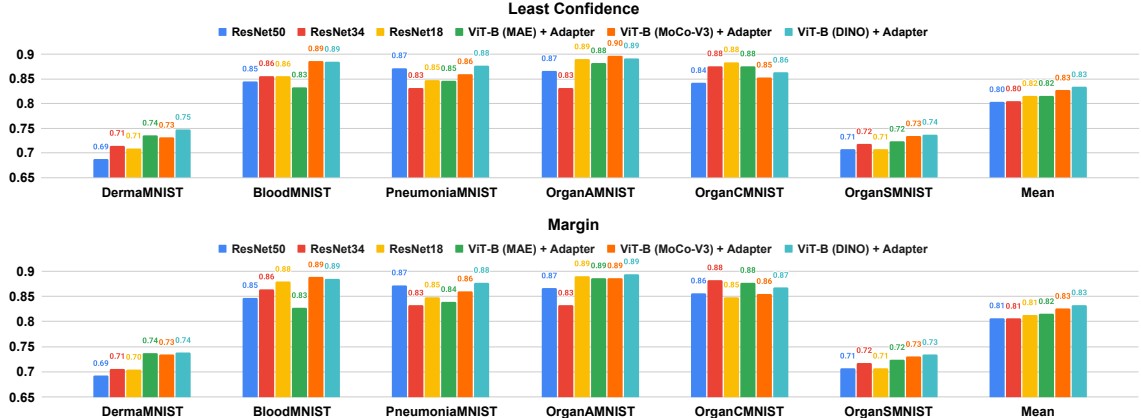

Figure 8: The AUBC results of comparing existing AL model structures (ResNet) and our efficient model structures (ViT-B + Adapter) via various AL strategies, including *Least Confidence* and *Margin* **Mean** here denotes the averaged values of results among 6 different datasets. The results are produced under the setting of Many-shot AL.

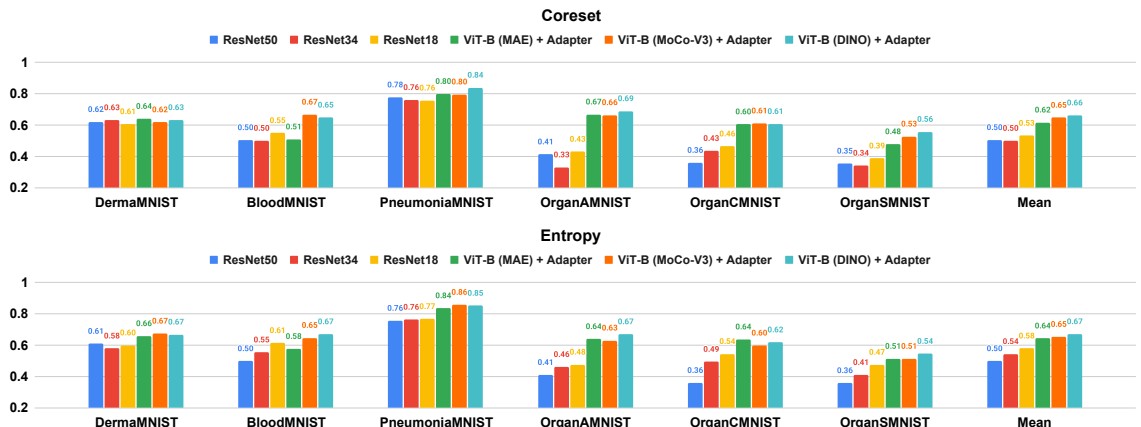

Figure 9: The AUBC results of comparing existing AL model structures (ResNet) and our efficient model structures (ViT-B + Adapter) via AL strategies as *Coreset* and *Entropy* under the setting of Few-shot AL. It can be viewed as the extension for Fig. 6.

| | DermaMNIST | BloodMNIST | PneumoniaMNIST | OrganAMNIST | OrganCMNIST | OrganSMNIST | **Mean** |
|---|---|---|---|---|---|---|---|
| Coreset | | | | | | | |
| ViT-B (MoCo-V3) | 0.6170 | 0.6660 | 0.7950 | 0.6630 | 0.6100 | 0.5260 | 0.6462 |
| + RandAug | 0.6520 | 0.6570 | 0.8170 | 0.6920 | 0.6130 | 0.5410 | 0.6620 |
| + AutoAug | 0.6290 | 0.6900 | **0.8390** | 0.6990 | 0.6170 | 0.5390 | 0.6688 |
| + NormalAug | 0.6280 | 0.6340 | 0.7720 | 0.5560 | 0.5800 | 0.5470 | 0.6195 |
| + SubstitutivePatchAug | **0.6610** | **0.7190** | 0.8260 | **0.7170** | **0.6860** | 0.5480 | **0.6928** |
| ViT-B (DINO) | 0.6310 | 0.6470 | 0.8380 | 0.6880 | 0.6060 | 0.5560 | 0.6610 |
| + RandAug | 0.6640 | 0.6440 | 0.8210 | 0.6790 | 0.6220 | 0.4700 | 0.6500 |
| + AutoAug | 0.6430 | 0.6500 | 0.8340 | **0.7200** | **0.6490** | 0.5410 | 0.6728 |
| + NormalAug | 0.6320 | 0.6720 | 0.8310 | 0.5570 | 0.5640 | 0.5170 | 0.6288 |
| + SubstitutivePatchAug | **0.7120** | **0.6940** | 0.8350 | 0.7140 | 0.6360 | **0.5940** | **0.6975** |
| Entropy | | | | | | | |
| ViT-B (MoCo-V3) | 0.6730 | 0.6460 | **0.8570** | 0.6290 | 0.5970 | 0.5120 | 0.6523 |
| + RandAug | 0.6560 | 0.6650 | 0.8320 | **0.6810** | 0.6130 | 0.5230 | 0.6617 |
| + AutoAug | 0.6460 | 0.6760 | 0.8420 | 0.6700 | 0.6140 | 0.5160 | 0.6607 |
| + NormalAug | 0.6550 | 0.6980 | 0.8240 | 0.5580 | 0.5540 | 0.5390 | 0.6380 |
| + SubstitutivePatchAug | **0.6860** | **0.6920** | 0.8240 | 0.6410 | **0.6450** | 0.5450 | **0.6722** |
| ViT-B (DINO) | 0.6670 | 0.6690 | 0.8510 | 0.6690 | 0.6170 | 0.5440 | 0.6695 |
| + RandAug | 0.6440 | **0.6960** | 0.8610 | **0.6800** | 0.6370 | 0.5240 | 0.6737 |
| + AutoAug | 0.6380 | 0.6590 | **0.8680** | 0.6720 | 0.6170 | 0.5270 | 0.6635 |
| + NormalAug | 0.6510 | 0.6790 | 0.8250 | 0.5430 | 0.5370 | 0.5450 | 0.6300 |
| + SubstitutivePatchAug | **0.6830** | 0.6920 | 0.8560 | 0.6660 | **0.6620** | 0.5550 | **0.6857** |

Table 7: More Results for the comparison study for DA methods under the setting of Few-shot AL. It can be viewed as the extension for Table 1.

| ViT-B (MAE) + Adapter | Least Confidence | | Margin | | LeastConfidence MC | | MeanSTD | |
|---|---|---|---|---|---|---|---|---|
| | With LaLS | Without LaLS | With LaLS | Without LaLS | With LaLS | Without LaLS | With LaLS | Without LaLS |
| BloodMNIST | **0.6270** | 0.6040 | **0.6470** | 0.5970 | **0.6200** | 0.5970 | 0.5650 | **0.5950** |
| DermaMNIST | **0.6920** | 0.6570 | **0.6880** | 0.6540 | **0.6570** | 0.6540 | 0.6260 | **0.6370** |
| PneumoniaMNIST | **0.8330** | 0.8230 | **0.8330** | 0.8230 | **0.8530** | 0.8230 | 0.8070 | **0.8320** |
| OrganAMNIST | 0.6840 | **0.6930** | 0.7220 | **0.7260** | 0.6750 | 0.7100 | 0.5680 | **0.6030** |
| OrganCMNIST | 0.6390 | **0.6530** | **0.7000** | 0.6810 | 0.6470 | **0.6740** | **0.6050** | 0.6040 |
| OrganSMNIST | 0.5170 | **0.5410** | **0.5650** | 0.5620 | 0.5230 | **0.5460** | **0.4620** | 0.4500 |
| Mean | **0.6653** | 0.6618 | **0.6925** | 0.6738 | 0.6625 | **0.6673** | 0.6055 | **0.6202** |

Table 8: Results of the Ablation Study on Instance-adaptive Label Smoothing (IaLS) conducted on ViT-B (MAE)

| | DermaMNIST | BloodMNIST | PneumoniaMNIST | OrganAMNIST | OrganCMNIST | OrganSMNIST | Mean |
|---|---|---|---|---|---|---|---|
| | Least Confidence | | | | | | |
| Adapter | **0.6513** | 0.5790 | **0.8350** | **0.6740** | 0.6420 | 0.5140 | **0.6492** |
| ResAdapter | 0.6263 | 0.5790 | 0.8210 | 0.6660 | **0.6570** | 0.5180 | 0.6446 |
| Adapter + SubstitutivePatchAug | **0.6700** | **0.6340** | 0.8280 | 0.6850 | **0.6710** | 0.5360 | **0.6707** |
| ResAdapter + SubstitutivePatchAug | 0.6410 | 0.6230 | 0.8160 | **0.7060** | 0.6570 | **0.5420** | 0.6642 |
| | Margin | | | | | | |
| Adapter | 0.6517 | **0.6020** | **0.8350** | 0.6980 | 0.6670 | 0.5400 | 0.6656 |
| ResAdapter | **0.6593** | 0.5970 | 0.8260 | 0.6790 | 0.6580 | 0.5320 | 0.6586 |
| Adapter + SubstitutivePatchAug | **0.6640** | 0.6410 | 0.8280 | 0.7240 | 0.7270 | **0.5850** | 0.6948 |
| ResAdapter + SubstitutivePatchAug | 0.6620 | **0.6490** | **0.8440** | **0.7330** | **0.7290** | 0.5830 | **0.7000** |

Table 9: Results for Different Model Structures of the Adapter $g$.

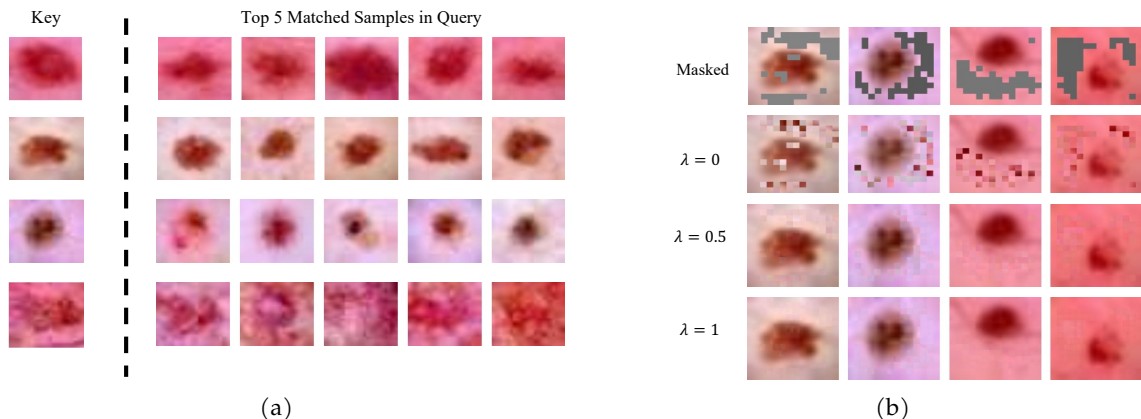

(a)                                    (b)

Figure 10: Fig. 10b:Visualization Results for the Query Set $Q$ when applying SubstitutivePatchAug. Fig. 10b: Visualization Results for SubstitutivePatchAug among different $\lambda$.

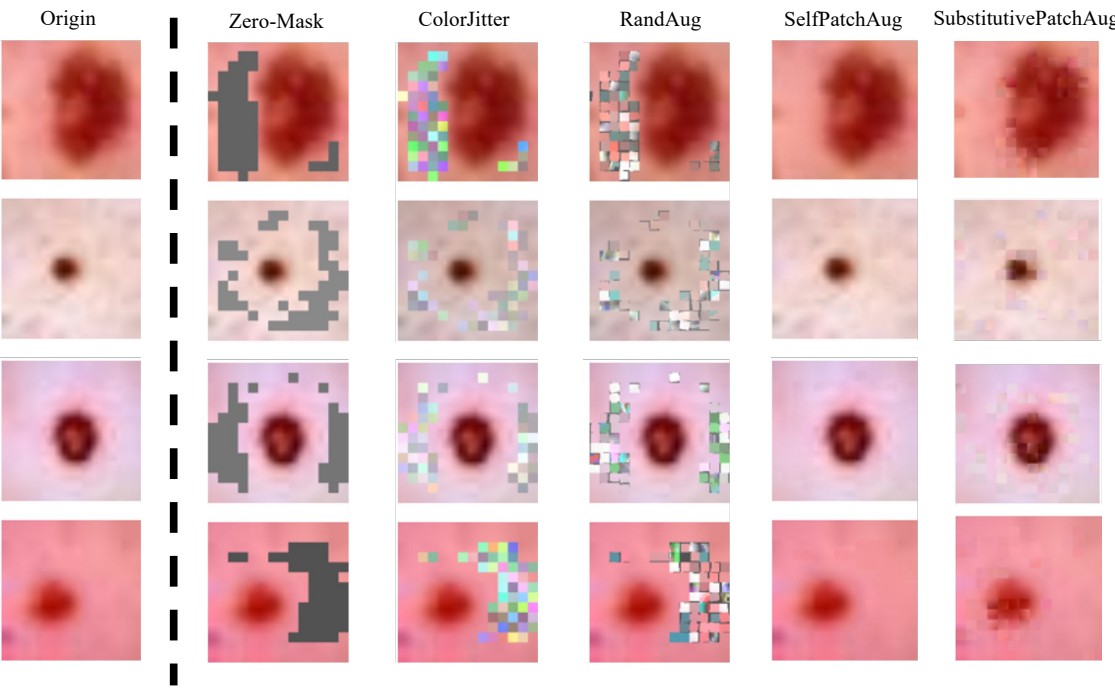

Figure 11: Visualization Results for Augmenting Label-irrelevant Patches with Different Methods

