# OpenReview forum: "Exploring Minimally Sufficient Representation in Active Learning through Label-Irrelevant Patch Augmentation"
_CPAL.cc/2024/Conference — CPAL 2024 (Proceedings Track) Oral_

### Official Review · Reviewer_VkB2 · 2023-10-07

**Rating:** 5
**Confidence:** 3

**Review:**

# Overview

The manuscript proposed a new framework to improve the performance of active learning, specifically for medical image classification.
The proposed method is based on a self-supervised model and a label-irrelevant patch augmentation scheme. Plenty of ablation studies have been conducted, showing the effectiveness of the proposed method.

# Comments

(+) Thorough ablation studies are carried out, providing a deep understanding of the method's effectiveness.

(+) The method is articulated clearly and supported with descriptive diagrams, enhancing comprehension.

(-) Regarding the results presented in Table 2, it's unclear if the FLOPs consider the overall costs or only account for individual rounds. Given that every AL round necessitates training a new adapter based on the current encoder, the overall computational overhead seems not negligible.

(-) The abstract mentioned, "to reduce redundancy in the learned features and mitigate overfitting during the AL process". Are there specific results that demonstrate the successful mitigation of overfitting?

(-) The advancements illustrated in Table 1 appear to be relatively mild. Incorporating error bars might offer a more effective assessment of the proposed method.

---

### Official Review · Reviewer_2XdG · 2023-10-08
**Review for Exploring Minimally Sufficient Representation in Active Learning through Label-Irrelevant Patch Augmentation**

**Rating:** 5
**Confidence:** 4

**Review:**

Summary Of The Paper

This paper studies active learning (AL) for medical image classification. The authors introduce an efficient AL framework based on self-supervised learning models. This paper also introduces a patch augmentation method to enhance the feature representation quality and reduce redundancy, ultimately preventing overfitting during AL.


Main Review

- Strength
1) The paper is written in a clear manner. The authors provide a comprehensive background and give clear justification and problem setting for their proposed work.

2) In addition, the proposed method is supported by multiple datasets, and the authors have conducted extensive ablation studies, including several versions of augmentations, which is good to justify the proposed model.

- Weakness
1) With extensive experiments and analysis, the paper has demonstrated its proposed strategy on several dataset, however, the intuition of the proposed method is not thoroughly discussed. I recommend the authors should elaborate more on the design of their label-irrelevant patches augmentations.

2) In order to demonstrate the effectiveness of the proposed method, downstream tasks such as detection on AL (e.g., https://github.com/yuantn/MI-AOD) could be explored. I think the proposed method may not be practical without other tasks.

3) Have the authors investigated and addressed potential domain discrepancies between the pretrained ViTs, which are trained on natural images, and their fine-tuned adaptor with medical images (specific domain)?

4) I'm not entirely certain if Table 2 provides an apples-to-apples comparison. In one case, the entire VIT backbone is frozen, and only the adapter is fine-tuned. However, in the other case, Resnet18-50 is trained end-to-end. What would happen if I were to take a pretrained Resnet18 and then add an adapter for fine-tuning?

5) I suggest the authors to provide a comprehensive survey on relevant works such as augmentation strategy on representation learning (including self-supervised learning), For example: [A] Kügelgen et al, Self-Supervised Learning with Data Augmentations Provably Isolates Content from Style, [B] Yeh et. al., SAGA: Self-Augmentation with Guided Attention for Representation Learning

Summary Of The Review

Overall, without discussing the intuition of the proposed method in details, I am not sure if the novelty that authors mention in the paper is reliable. Also, I think we need to see valid elaboration and the intuition of the proposed method, and scale up to other tasks. Combined with the weaknesses I mentioned above, I vote for 5. I would like to see authors response to consider raising the rating.

---

### Official Review · Reviewer_XeA1 · 2023-10-15
**Review for paper 8**

**Rating:** 5
**Confidence:** 4

**Review:**

The paper proposes an efficient AL framework based on off-the-shelf self-supervised learning models, and a label-irrelevant patch augmentation scheme is introduced to reduce redundancy in the learned features and mitigate overfitting in the progress of Active Learning. Some key positives and weaknesses are listed as:

Strength:
1. Their formulation of a label-irrelevant patch augmentation scheme that preserves semantic information is interesting.
2. The proposed parameter-efficient AL framework can boost the overall performance of Few-shot AL by 5% − 7%.
3. The paper is very well written and it includes many interesting ablation studies + experimental details (eg. aug analysis in the appendix).

Weakness:
1. I am concerned about the expense of training an adapter on top of a pre-trained and fixed ViT in every AL round. In what way, this additional training cost can compensate for the performance gain?
2. Performance benefits of the proposed approach seem very marginal and sometimes less than RandAug. Can authors provide some explanation?
3. Additional results on non-medical images on conventional CV datasets can provide more evidence of benefits. I am not sure why authors limited their evaluation to medical images. Some widely popular medical datasets like ChestX-rays etc. are missing.

---

### Meta-Review · Area_Chair_Lhk8 · 2023-11-13

**Recommendation:** Reject
**Confidence:** 5

**Metareview:**

In this paper, the authors address the challenge of obtaining ample labeled data for training deep learning models, particularly in the context of medical imaging, where data collection is costly and time-consuming. They focus on the concept of active learning (AL), which aims to maximize model performance using a limited number of labeled samples by iteratively expanding and labeling the training dataset. The primary objective of this study is to develop a feature representation that strikes a balance between sufficiency and minimality, thereby facilitating effective AL for medical image classification.

The authors propose an efficient AL framework that leverages readily available self-supervised learning models. Additionally, they introduce a patch augmentation scheme that they claim is insensitive to labels, aiming to reduce redundancy in the learned features and mitigate overfitting during the AL process. The authors also claim to validate the performance improvements achieved by their approach across various medical image classification tasks using different AL strategies. The reviewers thought some of the formulation and results were novel and interesting but raised various concerns including expense of training an adaptor, marginal benefits, limited experiments (even in the medical domain), lack of intuition, lack of precise comparisons. While the authors response addressed some of these concerns it is clear that many remain. All reviewers agree that the paper is marginally below the threshold and I concur and therefore cannot recommend acceptance at this time.

---

### Decision · Program_Chairs · 2023-11-19

**Decision:**

Accept (Oral)

**Comment:**

After a careful reconsideration and taking into account the reviewers' comments and concerns, we have decided to accept the paper to the conference.

The paper addresses a significant challenge in the field of medical image classification, namely, the need for ample labeled data for training deep learning models. The focus on active learning (AL) and the development of a feature representation that balances sufficiency and minimality is commendable.

The authors have proposed an efficient AL framework that leverages self-supervised learning models and introduced a patch augmentation scheme to enhance the AL process. While there were concerns raised by the reviewers, the authors' responses have addressed some of these concerns.

Upon further consideration, I believe that the paper presents novel and interesting formulations and results, even though some concerns remain. Given the potential value of the proposed approach and the authors' efforts to address the reviewers' comments, we recommend the paper for acceptance. However, we encourage the authors to continue refining their work and addressing the remaining concerns to further improve the paper's quality before the camera-ready.

The action PC chair for this paper is Qing Qu, who made the decision after carefully reading the paper as well as the comments by all reviewers and AC. The decision is agreed upon by all PC chairs.